# 1,25-Dihydroxyvitamin D Enhances the Regenerative Function of Lgr5^+^ Intestinal Stem Cells In Vitro and In Vivo

**DOI:** 10.3390/cells13171465

**Published:** 2024-08-31

**Authors:** Nisar Ali Shaikh, Chenfan Liu, Yue Yin, David J. Baylink, Xiaolei Tang

**Affiliations:** 1Department of Veterinary Biomedical Sciences, College of Veterinary Medicine, Long Island University, Brookville, NY 11548, USA; 2Shandong Public Health Clinical Center, Shandong University, Jinan 250013, China; 3Division of Regenerative Medicine, Department of Medicine, School of Medicine, Loma Linda University, Loma Linda, CA 92354, USA; 4Department of Basic Sciences, School of Medicine, Loma Linda University, Loma Linda, CA 92354, USA

**Keywords:** Lgr5^+^ intestinal stem cells, 1,25-dihydroxyvitamin D, vitamin D receptor, inflammatory bowel disease

## Abstract

Inflammatory bowel disease (IBD) is a chronic inflammatory disorder in the intestines without a cure. Current therapies suppress inflammation to prevent further intestinal damage. However, healing already damaged intestinal epithelia is still an unmet medical need. Under physiological conditions, Lgr5^+^ intestinal stem cells (ISCs) in the intestinal crypts replenish the epithelia every 3–5 days. Therefore, understanding the regulation of Lgr5^+^ ISCs is essential. Previous data suggest vitamin D signaling is essential to maintain normal Lgr5^+^ ISC function in vivo. Our recent data indicate that to execute its functions in the intestines optimally, 1,25(OH)_2_D requires high concentrations that, if present systemically, can cause hypercalcemia (i.e., blood calcium levels significantly higher than physiological levels), leading to severe consequences. Using 5-bromo-2′-deoxyuridine (BrdU) to label the actively proliferating ISCs, our previous data suggested that de novo synthesized locally high 1,25(OH)_2_D concentrations effectively enhanced the migration and differentiation of ISCs without causing hypercalcemia. However, although sparse in the crypts, other proliferating cells other than Lgr5^+^ ISCs could also be labeled with BrdU. This current study used high-purity Lgr5^+^ ISC lines and a mouse strain, in which Lgr5^+^ ISCs and their progeny could be specifically tracked, to investigate the effects of de novo synthesized locally high 1,25(OH)_2_D concentrations on Lgr5^+^ ISC function. Our data showed that 1,25(OH)_2_D at concentrations significantly higher than physiological levels augmented Lgr5^+^ ISC differentiation in vitro. In vivo, de novo synthesized locally high 1,25(OH)_2_D concentrations significantly elevated local 1α-hydroxylase expression, robustly suppressed experimental colitis, and promoted Lgr5^+^ ISC differentiation. For the first time, this study definitively demonstrated 1,25(OH)_2_D’s role in Lgr5^+^ ISCs, underpinning 1,25(OH)_2_D’s promise in IBD therapy.

## 1. Introduction

Inflammatory bowel disease (IBD) is a chronic inflammatory disorder of the gastrointestinal tract. It is estimated to affect approximately 1.3% (3 million) of adults in the United States (CDC data, accessed 16 May 2024). The incidence and prevalence continuously increase worldwide, making it a global health concern [1]. 

The disease is believed to be caused by abnormal immune responses to intestinal bacteria and manifests as two primary conditions: ulcerative colitis (UC) and Crohn’s disease (CD). Current management approaches block the abnormal immune responses, significantly slowing the disease progression and improving a patient’s quality of life. However, no cure is available. The disease progression is partly caused by continuous bacterial invasion into interior tissues due to damaged intestinal epithelia, which constantly stimulates the immune system and heightens inflammation [2,3]. Therefore, completely repairing damaged intestinal epithelia is critical for curing the disease. 

Intestinal epithelial repair depends on intestinal stem cells (ISCs) located in the intestinal crypts. Under physiological conditions, it is well accepted that Lgr5^+^ ISCs regenerate all the mature intestinal epithelia and replenish them every 3–5 days [4,5]. Therefore, understanding the regulation of Lgr5^+^ ISCs can potentially lead to novel strategies for promoting intestinal epithelial repair.

Previous data suggest that vitamin D signaling is essential to maintain normal Lgr5^+^ ISC function in vivo [6]. Our recent data indicate that to execute its functions in the intestines optimally, 1,25-dihydroxyvitamin D (1,25[OH]_2_D or the active vitamin D) requires high concentrations that, if present systemically, can cause hypercalcemia (i.e., blood calcium levels significantly higher than physiological levels), leading to severe consequences [7,8,9,10]. Our data demonstrated that de novo synthesized locally high 1,25(OH)_2_D concentrations robustly suppressed experimental colitis (an IBD animal model) without causing hypercalcemia [7,11]. Hence, de novo synthesized locally high 1,25(OH)_2_D concentrations represent a potential novel therapy for IBD [7,11].

The mechanisms underlying the suppression of colitis by de novo synthesized, locally elevated concentrations of 1,25(OH)_2_D can be complex. One mechanism may involve the actions of 1,25(OH)_2_D on the immune system. Studies, including ours, have demonstrated that high 1,25(OH)_2_D concentrations can suppress inflammation by directly promoting the generation of regulatory T cells [12,13,14,15,16,17,18] and inducing tolerance in antigen-presenting cells [19,20] within the immune system. Additionally, 1,25(OH)_2_D is known to protect the integrity of the intestinal epithelial barrier, [21,22,23,24] and its signaling is essential for maintaining the physiological function of Lgr5^+^ ISCs in mice [6].

Employing the pulse-and-chase strategy with 5-bromo-2′-deoxyuridine (BrdU) labeling, we recently showed that de novo synthesized locally high concentrations of 1,25(OH)_2_D promote the migration and differentiation of actively cycling cells within the intestinal crypts under both steady-state and inflammatory conditions [7]. Since other actively cycling cell subsets are present in the crypts alongside Lgr5^+^ stem cells [25,26,27,28,29], BrdU labeling did not selectively identify Lgr5^+^ ISCs.

Our current study utilized high-purity Lgr5^+^ ISC lines and a mouse model where Lgr5^+^ ISCs and their progeny could be tracked separately. This approach allowed us to investigate the effects of de novo synthesized locally high concentrations of 1,25(OH)_2_D on Lgr5^+^ ISC functions, which is fundamental for the regulation of intestinal repair and regeneration.

## 2. Materials and Methods

### 2.1. Animal Subjects

C57BL/6 (B6) male mice of 6–8 weeks of age were procured from The Jackson Laboratory (Bar Harbor, ME, USA) and housed in the Long Island University (LIU) post-campus animal facility. Before experimentation, the mice were allowed for at least 5 days of acclimation. Heterozygous Lgr5GFP-AI mice were generated in-house by crossing cre-inducible Lgr5-EGFP-IRES-creERT2 (Lgr5-GFP) mice (Jackson Laboratory) with the cre-reporter B6.Cg-Gt(ROSA)26Sor tm9(CAG-tdTomato)Hze/J (Ai9 or Ai) mice (Jackson Laboratory). The above transgenic mice were genotyped using Lgr5- and AI-specific primers (Appendix A).

The LIU Institutional Animal Care and Use Committee reviewed and approved all animal study protocols.

### 2.2. Crypt Single-Cell Isolation

Crypts were isolated following a previously described protocol with some modifications [5,30,31]. Initially, the proximal half of the colon was dissected and carefully washed with ice-cold PBS to remove luminal contents. The dissected intestine was then cut into 4–5 cm pieces. To invert the intestine and expose the luminal epithelia, we inserted a sterilized feeding needle into the intestine with the blunt tip in the front all the way to the distal end. Then, the distal end of the intestine and the needle’s blunt tip were tied with a suture. Subsequently, the needle was pulled towards the operator until all the intestinal epithelia had faced outside. The open end of the intestine was then tied with another suture to close the intestinal lumen.

The inverted intestine, with epithelia facing outside, was submerged in the ice-cold BD recovery solution within a 15 mL falcon tube, inflated with air, and left on ice for 1 h. During this time, the intestine was deflated and inflated every 5 min to facilitate the release of crypts. After 1 h, the intestine was removed, and the solution containing crypts was centrifuged at 300× *g* for 5 min. The pellet was then washed twice with ice-cold PBS.

To obtain single cells from the crypts, we first mixed the crypt pellet with a basal medium containing 500 U/mL Collagenase type IV by pipetting up and down using a 5 mL pipette and incubated in a 37 °C water bath for 30 min. Afterward, 10 mL of ice-cold PBS was added and vigorously pipetted ten times with a 10 mL pipette, and the supernatant fractions containing a high number of crypt cells were collected. In addition, the cells were pelleted, treated with a basal medium containing 15 U/mL DNase I at 37 °C for 5 min, filtered through a 70 μm cell strainer, and centrifuged at 300 g for 5 min. The resulting cell pellet was resuspended in a cell culture medium for high-purity Lgr5^+^ ISC or organoid cultures (see description below).

### 2.3. High-Purity Lgr5^+^ Single ISC Culture

The crypt single cells were cultured following a previously established protocol [5,30,31]. In brief, the cells were cultured in X-VIVO 15 serum-free media (Lonza Biosciences, Maryland, USA, Cat#: 04-418Q) supplemented with ENR growth factors (ENR culture medium), including EGF (50 ng/mL, PeproTech, Cranbury, NJ, USA, Cat#: 315-09-100UG), Noggin (100 ng/mL, PeproTech, Cranbury, NJ, USA, Cat#: 250-38-20UG), and R-spondin 1 (500 ng/mL, PeproTech, Cranbury, NJ, USA, Cat#: 315-32-50UG). In addition, the following chemicals (VC) were added to expand Lgr5^+^ ISCs selectively: valproic acid (1 mM, PeproTech, Cranbury, NJ, USA, Cat#: 1066656) and CHIR99021 (3 μM, PeproTech, Cranbury, NJ, USA, Cat#: 2520691). The cell culture medium (ENR+VC culture medium) was replenished every other day. 

The cells were passaged every 6 days at a 1:20 split ratio under the VC culture condition. Briefly, the cell culture medium was aspirated, and Accutase (Sigma-Aldrich, St. Louis, MA, USA, Cat#: A6964-500ML) was added. After being incubated at 37 °C for 10–20 min, the cells were dissociated by pipetting, washed, and plated into new wells in 24-well plates.

### 2.4. Organoid Culture

Crypt single cells were washed in ice-cold PBS, and pelleted cells were embedded in Matrigel (Corning, Corning, NY, USA, Cat#: 354234) and seeded on pre-warmed 48-well plates. After Matrigel solidification, the ENR culture medium was added.

The ENR culture medium was replenished every other day. The cells were passaged following the protocol previously described [32,33], with modifications. Briefly, Matrigel-embedded organoids were collected using a scraper into a 15 mL conical tube and incubated with Cell Recovery Solution (Corning, Corning, NY, USA Cat#: 354253) for 30 min on ice with continuous shaking to dissolve Matrigel. After centrifugation at 500× *g* for 5 min at 4 °C, organoids were washed in ice-cold PBS and centrifuged again. Next, the organoids were incubated with Accutase (Sigma-Aldrich, St. Louis, MA, USA, Cat#: A6964-500ML) at 37 °C for 10–20 min for dissociation into single cells. The solution was homogenized by pipetting, collected in a 15 mL conical tube, centrifuged at 500× *g* for 5 min at 4 °C, and washed twice in ice-cold PBS. Pelleted cells were embedded in new Matrigel and cultured as described above.

### 2.5. Generation of VDR Knockdown ISC Lines

We used the VDR Mouse shRNA Lentiviral Particle (Origene Technologies Inc., Rockville, MD, USA, Catalog# TL513148V) to knock down VDR in the Lgr5^+^ ISC lines. This plasmid contained a puromycin-resistant gene for selecting positively transduced cells. Before initiating the experiments, we determined the optimal puromycin concentration for killing non-transduced Lgr5^+^ ISCs. The minimal puromycin concentration that resulted in complete cell death was 1 µg/mL after 5 days.

To knock down VDR in the Lgr5^+^ ISCs, we added the lentiviral particles at a multiplicity of infection (MOI) of 20 into approximately 80% confluent cells in fresh culture medium containing polybrene (final concentration: 8 µg/mL). The cells were incubated at 37 °C, 5% CO_2_. After 24 h, the cell culture media were replenished. Puromycin was added after 48 h (about 2 days). Wells without the lentiviral particles were included as a control. After selection, VDR knockdown was confirmed by RT-PCR and Western blot.

### 2.6. Plasmid Constructs

Two lentiviral vectors were used to study the role of de novo synthesized locally high 1,25(OH)_2_D concentrations in Lgr5^+^ ISCs: the lenti-CYP (Cat#: Ecoli[VB220406-1373txq]) and the lenti-Ctr (Cat#: Ecoli[VB010000-9298rtf]). Both lentiviral vectors were obtained from VectorBuilder Inc, Chicago, IL, USA.

The lenti-CYP contained the human cytochrome P450 family 27 subfamily B member 1 gene (hCYP27B1, NM_000785.4) that encoded the 1,25(OH)_2_D-synthesizing enzyme. Hence, cells transduced with the lenti-CYP viruses de novo synthesized high 1,25(OH)_2_D concentrations. Briefly, the complementary CYP27B1 cDNA, which was 1.6 kilobases (kb) in length, was engineered to include a 5′ Kozak ribosome entry sequence. This modified cDNA was then incorporated into the pRRE.cPPT.PGKPuro.WPRE lentiviral vector (pLV[Exp]-Puro-SFFV>hCYP27B1), in which the hCYP27B1 expression was driven by the spleen focus-forming virus (SFFV) promoter. In addition, the vector also contained a puromycin resistance gene that was regulated by the phosphoglycerate kinase (PGK) promoter (Appendix A).

The control lentiviral vector (lenti-Ctr) contained GFP and puromycin resistance genes driven by the cytomegalovirus (CMV) promoter and mCherry gene by the EF1A promoter (pLV[Exp]-EGFP:T2A:Puro-EF1A>mCherry) (Appendix A).

### 2.7. Preparation of Lentiviruses

Lentiviruses were produced, as described previously [7,18]. To generate lentiviruses carrying one of the transfer plasmids (lenti-Ctr or lenti-CYP), we cultured HEK293T cells in the complete DMEM medium containing 10% FBS, 100 U/mL penicillin/streptomycin, 0.05 mM 2-ME, 1 mM sodium pyruvate, 0.1 mM nonessential amino acids, and 2 mM L-glutamine. The culture medium was replenished when the cells reached 70–80% confluence. A transfection solution prepared in Opti-MEM reduced serum medium (Gibco, Waltham, MA, USA, Cat#: 31985070), which contained lipofectamine 3000 (Invitrogen, Waltham, MA, USA, Cat#: L3000015), PMD2G (VSVG), pCMVR8.74 (Capsid), and the transfer plasmid, was added dropwise to the cells. The cells were then incubated at 37 °C and 5% CO_2_ for 24 h; after which, the transfection solution was substituted with DMEM medium containing 4% FBS, 100 U/mL penicillin/streptomycin, and 20 mM HEPES. Following further incubation at 37 °C and 5% CO_2_ for 48 h, the supernatants were collected, filtered through a 0.45 μm syringe filter (Millipore, Burlington, MA, USA, Cat#: SLHVR33RS), and centrifuged at 90,000× *g* at 4 °C for 90 min. The virus pellet was reconstituted in PBS containing 5% glycerol and titrated using puromycin-based limiting dilution. The typical viral titer ranged from 10^8^ to 10^9^ transducing units/mL.

### 2.8. Generation of Inflammation- and Gut-Homing Macrophages (CD11b^+^Gr1^+^) (MAC Cells) from Bone Marrow and Lentivirus Transduction

In this study, we transduced the MAC cells with the lenti-CYP viruses to generate MAC-CYP cells that de novo synthesized local high 1,25(OH)_2_D concentration. In addition, we transduced the MAC cells with the lenti-Ctr viruses to produce MAC-Ctr cells that served as a control. The protocols for generating the MAC-CYP and MAC-Ctr cells were described in our previous studies [7]. Briefly, after red blood cells were lysed, bone marrow cells were subjected to CD11b^+^Gr1^+^ monocyte purification using magnet beads. Specifically, Gr1^+^ cells were isolated using biotin-conjugated anti-Gr1 antibody (Miltenyi Biotec, Gaithersburg, MD, USA, Cat#: 130-120-829) and anti-biotin microbeads (Miltenyi Biotec, Gaithersburg, MD, USA, Cat#: 130-090-485). These Gr1^+^ cells were then cultured in X-VIVO 15 serum-free media (Lonza Biosciences, Maryland, USA, Cat#: 04-418Q) supplemented with recombinant murine GM-CSF (PeproTech, Cranbury, NJ, USA, Cat#: 315-03), recombinant murine G-CSF (PeproTech, Cat#: 250-05), and recombinant murine IL-13 (PeproTech, Cranbury, NJ, USA, Cat#: 210-13), each at a concentration of 100 ng/mL, for a period of seven days. After seven days, the cells were transduced with lenti-CYP or lenti-Ctr viruses at a multiplicity of infection (MOI) of 400. Following a 24 h incubation, the virus was removed, and the culture media were replenished with the addition of puromycin (0.5 μg/mL, Sigma-Aldrich St. Louis, MA, USA, Cat#: P4512). The cells were further cultured for 48 h and assessed for transduction efficiency using fluorescence microscopy. If necessary, the transduction process was repeated once before utilization.

### 2.9. Experiments for Tracking Lgr5^+^ ISCs In Vivo

Lgr5GFP-AI mice received an intraperitoneal injection of tamoxifen (75 mg/kg body weight) (Tamx) to permanently mark the Lgr5^+^ ISCs and their progeny with tdTomato one day before the treatment. The following day, the mice were administered 2 × 10^6^ cells/mouse of MAC-Ctr or MAC-CYP cells in 100 μL PBS. Five days later, intestines were harvested and subjected to either paraffin embedding for H&E staining or cryo-sectioning for confocal analysis. The mice did not receive the 1α-hydroxylase substrate, i.e., 25(OH)D3.

### 2.10. Experiments Using DC-CYP and DC-Ctr Cells

DC 2.4 cells, kindly provided by Dr. Kenneth L. Rock [34], are a DC line generated from bone-marrow-derived DCs and have been shown to recapitulate DC functions in vivo closely [35,36]. DC 2.4 cells were cultured in X-VIVO 15 serum-free media (Lonza Biosciences, Maryland, USA, Cat#: 04-418Q) at 5% CO_2_ and 37 °C. At 70% confluency, the DC 2.4 cells were transduced with lenti-CYP or lenti-Ctr (multiplicity of infection = 40), as described above, to generate DC-CYP and DC-Ctr cells, respectively. 

For the in vivo experiment, 2 × 10^6^ DC-CYP or DC-Ctr cells/mouse in 100 μL PBS were administered intraperitoneally. Five days later, intestines were harvested and subjected to either paraffin embedding for H&E staining or cryo-sectioning for confocal analysis.

### 2.11. Induction and Treatment of Experimental Colitis

To induce experimental colitis, B6 mice were provided with 3% dextran sodium sulfate (DSS) (MP Biomedicals, molecular weight = 36–50 kDa, OH, USA) in their drinking water for 7 days ad libitum. Body weight and mortality were recorded daily. On day 7, the mice received an intraperitoneal injection of 2 × 10^6^ cells/mouse of MAC-Ctr or MAC-CYP cells. A separate group of healthy mice was included as a control. On day 14, distal inflamed colons were collected for analysis. The mice did not receive the 1α-hydroxylase substrate, i.e., 25(OH)D3.

### 2.12. Histological Analysis

The intestines were dissected to remove adipose tissue and cut longitudinally. The luminal content was cleared by rinsing the tissue in cold phosphate-buffered saline (PBS). Starting from the distal end (rectum) and with the luminal side facing up, the intestines were rolled into a Swiss-roll configuration, positioning the distal colon at the center and the proximal colon on the outer side. The Swiss roll was placed in a histology plastic cassette and snap-frozen for one minute in a bath of liquid nitrogen-cooled isopentane. The frozen tissues were then embedded in Optimal Cutting Temperature compound (OCT, Sakura Tissue-TEK) on dry ice and stored at −80 °C. OCT blocks were sectioned using a pre-cooled cryostat at a thickness of 10 µm. Tissue sections were analyzed by Zeiss LSM 900 (Carl Zeiss Inc., Jena, Germany) confocal fluorescence microscope to identify GFP^+^ Lgr5^+^ stem cells and tdTomato^+^ mature intestinal epithelial cells. In addition, the tissues were also embedded in paraffin, fixed, and stained with hematoxylin and eosin (H&E). Imaging was performed using a Zeiss LSM 900 (Carl Zeiss Inc., Jena, Germany) confocal fluorescence microscope at 10× and 63× magnifications. 

### 2.13. Immunocytochemistry Analysis

For nonadherent cells, 1 × 10^6^ cells/mL were fixed with 4% paraformaldehyde in the culture media for 20 min at room temperature. The fixed cell solution (1 mL) was transferred to a 1.5 mL microfuge tube. After a 30 s centrifugation, the supernatant was discarded, and the cell pellet was resuspended in 1 mL of deionized water. This centrifugation and resuspension process was repeated twice. Finally, the pellet was resuspended in 200 µL of deionized water. A 5 µL aliquot of this cell suspension was used to create smears on a gelatin-coated slide 2% (*w*/*v*). The slide was dried on a hot plate with a low heat setting.

For adherent cells, the coverslips were sterilized by dipping them in ethanol, flaming, or exposing them to UV radiation for 1 h in a tissue culture hood. The cells were then seeded onto the sterile coverslips and allowed to grow to semi-confluency. 

The slides and coverslips with cells from the above were then fixed at room temperature for 20 min using 4% paraformaldehyde and washed three times with PBS. Subsequently, the cells were permeabilized with 0.5% Triton X-100 in PBS for 20 min, washed three times in PBST (PBS with 0.1% tween 20), and treated with PBS containing 3% BSA for 30 min to block non-specific binding. Then, the cells were incubated overnight at 4 °C with one of the following primary rabbit anti-mouse antibodies diluted in PBS containing 3% BSA: anti-intestinal alkaline phosphatase (Thermo Fisher, Waltham, MA, USA, Cat#: PA5-22210, 1:500), anti-Muc2 (Proteintech, Rosemont, IL, USA, Cat#: 27675-1-AP, 1:500), anti-chromogranin A (Proteintech, Rosemont, IL, USA, Cat#: 10529-1-AP, 1:500), and anti-lysozyme (Proteintech, Rosemont, IL, USA, Cat#: 15013-1-AP, 1:300). After the primary antibody incubation, the cells were washed three times with PBST and exposed to Alexa Fluor Plus 488-conjugated goat anti-rabbit antibody (Invitrogen, Waltham, MA, USA, Cat#: A48282, 1:1000) prepared in PBS containing 3% BSA for 1 h at room temperature, followed by three washes with PBST. Confocal microscopy (Zeiss LSM 900) was used to acquire images at 63× magnification.

### 2.14. Flow Cytometry Analysis

We set up different panels in advance using different cells in our laboratory flow cytometer. For each new analysis, we imported a relevant panel, which was then further adjusted using two controls: isotype antibodies and fluorescence minus one (FMO).

To characterize the Lgr5^+^ ISCs, we conducted a multi-color flow cytometry analysis, as described in our previous publications [7,18]. In brief, approximately 0.5~1.0 × 10^6^ cells were washed with PBS (pH 7.3) and stained with Zombie Aqua fixable viable dye (BioLegend, San Diego, CA, USA, Cat#: 423101). Lgr5 stem cells were cell-surface stained with the DyLight 488 fluorochrome-conjugated anti-Lgr5 (clone OTI2A2; Origene Technologies Inc., Rockville MD, USA, Cat# TA400002) at 4 °C for 30 min and washed twice with the FACS buffer. Subsequently, cells were fixed and permeabilized using the FoxP3/Transcription Factor Staining Buffer Set (eBioscience, San Diego, CA, USA, Cat#: 00-5523-00) and stained with FITC-conjugated anti-Bcl-2 (clone 10C4, Invitrogen, Cat#: 11-6992-41), PE/Cyanine7-conjugated anti-Ki-67 (clone 16A8, BioLegend, San Diego, CA, USA, Cat#: 652425) at 4 °C for 30 min. 

We performed single-color flow cytometry analysis to analyze the expression of mature intestinal epithelial markers. Briefly, cells were fixed and permeabilized using the FoxP3/Transcription Factor Staining Buffer Set (eBioscience, San Diego, CA, USA, Cat#: 00-5523-00). The cells were first stained with one of the following primary rabbit anti-mouse antibodies at 4 °C for 30 min: anti-intestinal alkaline phosphatase (Thermo Fisher, Waltham, MA, USA, Cat#: PA5-22210, 1:500), anti-Muc2 (Proteintech, Rosemont, IL, USA, Cat#: 27675-1-AP, 1:500), anti-chromogranin A (Proteintech, Rosemont, IL, USA, Cat#: 10529-1-AP, 1:500), and anti-lysozyme (Proteintech, Rosemont, IL, USA, Cat#: 15013-1-AP, 1:300). After the primary antibody incubation, the cells were washed three times with FACS buffer (PBS with 1% FBS and 0.05% sodium azide). Subsequently, the cells were exposed to Alexa Fluor Plus 488-conjugated goat anti-rabbit antibody (Invitrogen, Waltham, MA, USA, Cat#: A48282, 1:1000) at room temperature for 30 min, followed by three washes with the FACS buffer.

The above-stained cells were analyzed using Cytek Northern Lights flow cytometer (Cytek, San Diego, CA, USA). Data analysis was performed using FCS Express Software version 7.18.0021 (De Novo Software, Pasadena, CA, USA).

### 2.15. Quantitative Real-Time RT-qPCR

Cell and tissue mRNA was extracted using the RNeasy Mini Kit (Qiagen, Valencia, CA). cDNA was then synthesized using the High-Capacity cDNA Reverse Transcription Kit along with RNase inhibitor (Thermo Fisher). Subsequently, real-time RT-qPCR analysis was conducted using the SYBR Green PCR Master Mix (Applied Biosystems, Carlsbad, CA, USA) on the 7500 Fast Real-Time RT-qPCR System (Applied Biosystems). Data were normalized to GAPDH as an internal reference and presented as fold change relative to control samples. The primers for each target gene are provided in Appendix A.

### 2.16. Western Blot Analysis

The tissues, snap-frozen in liquid nitrogen, and cell pellets were lysed in RIPA buffer supplemented with protease and phosphatase inhibitors. Following lysis, the protein extracts were obtained by centrifugation. The protein concentration in each sample was determined using the BCA assay. Subsequently, an SDS-PAGE gel was prepared with an appropriate percentage based on the target protein size. Equal amounts of protein samples, along with molecular weight markers, were loaded onto the gel. Electrophoresis was performed until the desired separation of proteins was achieved. Next, protein transfer was carried out using a wet transfer system, transferring the proteins from the gel to a PVDF membrane. The transferred membrane was then blocked using a blocking buffer to prevent non-specific binding. Primary antibodies specific to the target proteins were diluted in an antibody diluent and incubated with the membrane overnight at 4 °C. After washing, the membrane was incubated with secondary antibodies conjugated to HRP at room temperature for 1 h. Protein bands were visualized using chemiluminescent substrate-conjugated secondary antibodies. Image J analysis software was utilized to quantify the protein bands, which were then normalized to suitable loading controls. Statistical analysis was conducted to determine the significance of any observed differences between experimental groups.

The following primary antibodies were used: anti-intestinal alkaline phosphatase (Thermo Fisher, Waltham, MA, USA, Cat#: PA5-22210, 1:1000), anti-Muc2 (Proteintech, Rosemont, IL, USA, Cat#: 27675-1-AP, 1:1000), anti-chromogranin A (Proteintech, Rosemont, IL, USA, Cat#: 10529-1-AP, 1:1000), and anti-lysozyme (Proteintech, Rosemont, IL, USA, Cat#: 15013-1-AP, 1:300). HRP-conjugated goat anti-rabbit IgG (H+L) cross-adsorbed (Thermo Fisher, Waltham, MA, USA, Cat#: G-21234, 1:5000) was used as secondary antibody.

Western blots were quantified by measuring the average intensity of the protein bands using ImageJ software. The intensity of the proteins of interest was then normalized to the intensity of housekeeping protein bands. 

### 2.17. Statistical Analysis

Statistical analyses were conducted using GraphPad Prism 9 software (GraphPad Software). The normality of the data was assessed, followed by either parametric or non-parametric statistics based on the results. ANOVA tests were utilized for experiments with more than two groups, while Student’s *t*-tests were employed for experiments with only two groups. Data were presented as means ± SEM, and results were considered statistically significant when the *p*-value was less than 0.05.

## 3. Results

### 3.1. High-Purity Lgr5^+^ ISC Lines Express Stemness Markers and Molecules Necessary for 1,25(OH)_2_D Signaling

To study 1,25(OH)_2_D’s effects on Lgr5^+^ ISCs, we generated high-purity Lgr5^+^ ISC lines by supplementing the cultures with valproic acid (V, a histone deacetylase inhibitor) and CHIR99021 (C, a glycogen synthase kinase 3β inhibitor) in addition to the ENR (epidermal growth factor [EGF], Noggin, and R-spondin 1) that is used to culture intestinal organoids, as described previously [37]. Flow cytometry analysis confirmed that 99.44% of the cells expressed Lgr5 (Figure 1A). Additionally, more than 99.72% of the cells were Ki-67^+^, consistent with the active cycling nature of Lgr5^+^ ISCs [4]. Approximately 75.47% of the cells contained Bcl-2, an anti-apoptotic molecule, explaining their ability to proliferate in cultures for a long period of time. Moreover, real-time RT-qPCR analysis showed that the cell lines also expressed Ascl2 and Smoc2 stemness markers (Figure 1B) [38,39]. Hence, the Lgr5^+^ ISC lines maintained in the ENR-VC condition are bonafide stem cells, which closely mimic the intestinal crypt microenvironment.

To determine the potential response of the Lgr5^+^ ISC lines to 1,25(OH)_2_D, we examined the molecules associated with vitamin D metabolism by real-time RT-qPCR. Our data demonstrated that the cells expressed the receptors necessary for 1,25(OH)_2_D’s genetic action, i.e., the vitamin D receptor (VDR) and the retinoid X receptor (RXR) (Figure 1C). The RXR has three subtypes (RXRα, RXRβ, and RXRγ), and the Lgr5^+^ ISCs expressed RXRα and RXRβ but not RXRγ. Considering this, the cells also expressed 1,25(OH)_2_D-synthesizing (1α-hydroxylase encoded by CYP27B1) and degrading (24-hydroxylase encoded by CYP24a1) enzymes, suggesting their ability to modulate vitamin D metabolism locally.

### 3.2. High-Purity Lgr5^+^ ISC Lines Initiate Differentiation after Removing Chemical Inhibitors (Valproic Acid and CHIR99021)

Previous data suggested that removing valproic acid and CHIR99021 initiated Lgr5^+^ ISC differentiation [37]. Since our Lgr5^+^ ISC lines were cultured in serum-free media, which differed from the previous study, it was necessary to know whether the cells would differentiate without the two inhibitors under our unique culture condition. Serum-free media were used because sera contained high concentrations of 25-hydroxy vitamin D (the 1α-hydroxylase substrate), and the Lgr5^+^ ISCs expressed 1α-hydroxylase (Figure 1C), which could complicate data interpretation.

We observed that the cell morphology changed after the valproic acid and CHIR99021 were removed from the cell cultures for one week (Figure 2A). Consistent with the morphology change, the mRNA expression of mature intestinal epithelial markers significantly increased one and two weeks after valproic acid and CHIR99021 removal, including Alpi (alkaline phosphatase, intestine; a marker for enterocytes [40]), chromogranin A (Chga; a marker for enteroendocrine cells [41]), Mucin 2 (Muc2; a marker for Goblet cells [42]), and lysozyme (Lyz1; a marker for Paneth cells [43]) (Figure 2B). Western blot analysis showed that the protein expression of the above markers gradually increased three, five, and seven days after the inhibitor removal (Figure 2C), further confirmed by immunocytochemistry (Figure 2D,E). Hence, the culture condition without VC recapitulates the intestinal microenvironment outside the crypts. In conclusion, the high-purity Lgr5^+^ cell lines are invaluable tools for understanding 1,25(OH)_2_D’s effects on Lgr5^+^ ISCs and their differentiation.

### 3.3. The 1,25(OH)_2_D at Concentrations Higher than Physiological Levels Strongly Enhances the Gene and Protein Expression of Mature Intestinal Epithelial Markers in the Lgr5^+^ ISC Lines in the Presence and Absence of Valproic Acid and CHIR99021

Our previous BrdU pulse-and-chase studies demonstrate that locally high 1,25(OH)_2_D concentrations significantly enhance the regenerative function of the actively cycling crypt cells, which include the Lgr5^+^ and other ISCs. Hence, we cultured the high-purity Lgr5^+^ ISC lines with different 1,25(OH)_2_D concentrations (0, 0.1, 10, and 100 nM) in the continuous presence of valproic acid and CHIR99021. The purpose was to cast light on how 1,25(OH)_2_D impacts Lgr5^+^ ISCs inside the crypts. On days 3, 5, and 7, the cells were analyzed by real-time RT-qPCR, Western blot, and flow cytometry. Our data showed that 1,25(OH)_2_D more strongly enhanced the mRNA (Figure 3A; real-time RT-qPCR) and protein (Figure 3B,C; Western blot and flow cytometry) expression of Alpi, Chga, Muc2, and Lyz1 at high than low concentrations. Despite the increased expression of mature intestinal epithelial markers, the expression of the genes associated with stemness were not significantly decreased (Appendix A), suggesting 1,25(OH)_2_D promotes Lgr5^+^ ISC differentiation as well as maintains the stemness of these cells.

We also determined how 1,25(OH)_2_D affects the Lgr5^+^ ISCs upon removing valproic acid and CHIR99021 to initiate differentiation. Similarly, 1,25(OH)_2_D more robustly enhanced the mRNA (Figure 4A; real-time RT-qPCR) and protein (Figure 4B,C; Western blot and flow cytometry) expression of Alpi, Chga, Muc2, and Lyz1 at high than low concentrations. Under this condition, the mRNA expression of Lgr5 and Ascl2 were significantly reduced, whereas proliferation remained unchanged, as indicated by the stable Ki-67 expression (Appendix A). These data are consistent with Lgr5 downregulation after crypt stem cells migrate out of intestinal crypts [4].

It is worth mentioning that the high 1,25(OH)_2_D concentrations were significantly higher than the physiological blood levels (approximately 38~134 pM) [18,44]. Hence, the current data further support our previous findings on 1,25(OH)_2_D’s biological actions [7,11,16,18,45,46].

### 3.4. The 1,25(OH)_2_D at Concentrations Higher than Physiological Levels Vigorously Augments Mature Intestinal Epithelial Markers’ Gene and Protein Expression in Colon Organoid Cultures

The above studies were performed using high-purity Lgr5^+^ ISC lines. To mimic in vivo conditions more closely, we grew organoids from intestinal crypt cells in Matrigel under the ENR culture condition (Figure 5A). The organoids have crypts that contain ISCs and villi that contain mature intestinal epithelial cells, recapitulating the complete crypt-villus structure in the intestines. Similar to the findings using high-purity Lgr5^+^ ISC lines, 1,25(OH)_2_D more strongly enhanced the mRNA (Figure 5B; real-time RT-qPCR) and protein (Figure 5C; Western blot) expression of Alpi, Chga, Muc2, and Lyz1 at high than low concentrations. In addition, despite the enhanced differentiation, 1,25(OH)_2_D at high concentrations did not suppress stemness, proliferation, and survivability (Figure 5D).

### 3.5. The 1,25(OH)_2_D’s Functions in Lgr5^+^ ISCs are Primarily Mediated by VDR

Although 1,25(OH)_2_D’s genomic actions mediated by VDR appear primary, non-genomic actions have been reported. In this regard, 1,25(OH)_2_D has been shown to bind a cell membrane receptor, i.e., protein disulfide isomerase family A, member 3 (Pdia3), to produce non-genomic actions, which can also regulate gene expression [47,48]. To evaluate the contribution of genomic VDR to the effects of 1,25(OH)_2_D on the Lgr5^+^ ISCs, we knocked down VDR using a commercial shRNA kit. Then, we added different 1,25(OH)_2_D concentrations (0, 0.1, 10, and 100 nM) to wild-type and VDR knockdown Lgr5^+^ ISCs in the presence of valproic acid and CHIR99021. Our data showed that VDR knockdown significantly reduced the mRNA (Figure 6A) and protein (Figure 6B,C) expression of Alpi, Chga, Muc2, and Lyz1. In addition, we performed a similar experiment on the wild-type and VDR knockdown Lgr5^+^ ISCs in the absence of valproic acid and CHIR99021 and obtained similar results (Figure 7). Hence, genomic VDR plays a major role in mediating 1,25(OH)_2_D’s functions in Lgr5^+^ ISCs. Interestingly, we noticed that the mRNA and protein expression of Alpi, Chga, Muc2, and Lyz1 in the VDR-deficient cells were also decreased in the absence of exogenous 1,25(OH)_2_D treatment (Figure 6 and Figure 7). The data support a previous report showing that VDR has ligand-independent functions [49].

### 3.6. De Novo Synthesized Locally High 1,25(OH)_2_D Concentrations Modulate Local Vitamin D Metabolism

So far, our data further suggest that 1,25(OH)_2_D requires concentrations significantly higher than physiological levels to execute its functions efficiently in Lgr5^+^ ISCs. To circumvent hypercalcemia associated with systemic high 1,25(OH)_2_D concentrations, we previously showed that inflammation- and gut-homing macrophages (MAC cells) engineered to overexpress the 1α-hydroxylase (MAC-CYP cells) robustly suppressed experimental colitis without the hypercalcemia [7,11]. However, our previously published studies only posited that the MAC-CYP cells could upregulate 1α-hydroxylase levels locally without substantial data [7,11]. Hence, we decided to understand how the MAC-CYP cells modulated local vitamin D metabolism in inflamed intestines in this study.

Accordingly, B6 mice were induced for DSS colitis and received different treatments upon disease onset (Figure 8A). The data confirmed our previous findings that the MAC-CYP cells but not the control treatments effectively suppressed the experimental colitis, as demonstrated by body weight (Figure 8B) and colon length (Figure 8C,D). The current data further demonstrated that inflammation significantly increased the mouse CYP27B1 and VDR mRNA expression in the inflamed distal colons (Figure 8E). Since the MAC-CYP cells contain the human CYP27B1 gene, we measured human CYP27B1 mRNA expression. Our data showed that the human CYP27B1 mRNA expression was significantly augmented by the MAC-CYP cells but not the control treatments (Figure 8F). Western blot analysis using an anti-CYP27B1 antibody cross-reactive with mouse and human CYP27B1 showed that the MAC-CYP cell treatment significantly enhanced the protein expression of total CYP27B1 (Figure 8G,H). Therefore, our data support that the MAC-CYP cells significantly increase local 1,25(OH)_2_D synthesis to effectively execute 1,25(OH)_2_D functions without causing hypercalcemia (Appendix A).

### 3.7. De Novo Synthesized Locally High 1,25(OH)_2_D Concentrations Enhance Lgr5^+^ ISC Differentiation In Vivo

To precisely determine MAC-CYP cells’ effects on Lgr5^+^ ISCs, we generated a mouse strain (Lgr5GFP-AI mice) in which GFP expression was dependent on Lgr5 expression, but tdTomato was expressed in Lgr5^+^ ISCs and their progeny permanently independent of Lgr5 expression upon tamoxifen administration. Because most mature intestinal epithelial cells lose Lgr5 expression under physiological conditions, GFP is predominantly only located in the intestinal crypts. In contrast, tdTomato is independent of Lgr5 expression, and hence, red color is present in both intestinal crypts and villi. We intraperitoneally injected the Lgr5GFP-AI mice with tamoxifen once to activate tdTomato expression. The mice were treated with MAC-CYP or control cells on the second day for five days. We confirmed the mice did not have hypercalcemia (Appendix A). In addition, our data demonstrated that the MAC-CYP cells significantly increased tdTomato^+^ cells along the intestinal villi, suggesting enhanced migration and differentiation of the Lgr5^+^ ISCs (Figure 9A–E). Additionally, data from the experiment using dendritic cells engineered to overexpress CYP27B1 (DC-CYP cells) displayed similar results (Appendix A). Hence, our data further consolidate our previous findings from the BrdU pulse-and-chase analysis.[7]

## 4. Discussion

One hallmark of IBD is the extensive damage to the epithelial layers [50], which allows continuous invasion of intestinal luminal bacteria into interior tissues to stimulate the immune system and worsen chronic inflammation. Recent data suggest that 1,25(OH)_2_D has several functions that can alleviate this pathogenic mechanism. First, it can suppress chronic immune responses by directly generating regulatory T cells [12,13,14,15,16,17,18] and inducing tolerance in antigen-presenting cells in the immune system [19,20]. Second, it can protect the integrity of intestinal epithelial barrier function [21,22,23,24]. Third, it stimulates the secretion of antimicrobial molecules [51,52], which can help reduce the bacteria load in tissues. Data from this study further demonstrate that it can also enhance Lgr5^+^ ISC function to promote epithelial regeneration and repair, underscoring the importance of 1,25(OH)_2_D in IBD treatment.

In this study, we employed two strategies to assess the role of 1,25(OH)_2_D in Lgr5^+^ ISCs. In one strategy, we generated high-purity Lgr5^+^ ISC lines and assessed their differentiation in vitro into mature intestinal epithelial cells using well-defined markers, i.e., Alpi for mature enterocytes, [40] Chga for enteroendocrine cells, [41] Muc2 for Goblet cells [42], and Lyz1 for Paneth cells [43]. Our data demonstrated that 1,25(OH)_2_D at concentrations significantly higher than physiological levels enhanced Lgr5^+^ ISC differentiation in vitro (Figure 1, Figure 2, Figure 3, Figure 4, Figure 5, Figure 6 and Figure 7). 

In another strategy, we bred a mouse strain in which Lgr5^+^ ISCs and their progeny could be tracked separately in vivo (Figure 9). We showed that de novo synthesized locally high 1,25(OH)_2_D concentrations via the MAC-CYP cells augmented Lgr5^+^ ISC differentiation. To our knowledge, this study, for the first time, definitively demonstrates the pro-differentiating role of 1,25(OH)_2_D in Lgr5^+^ ISCs.

Despite the promising functions of 1,25(OH)_2_D, its clinical application is far from satisfactory [53,54,55,56]. We have proposed that 1,25(OH)_2_D’s hypercalcemia effect is one primary obstacle because concentrations significantly higher than physiological levels are necessary for 1,25(OH)_2_D to effectively execute some of its newly discovered biological functions [7,11,16,18,45,46,57]. Our current data from the high-purity Lgr5^+^ ISC lines show that 1,25(OH)_2_D requires concentrations significantly higher than the physiological levels to enhance Lgr5^+^ ISC differentiation optimally in vitro (Figure 1, Figure 2, Figure 3, Figure 4, Figure 5, Figure 6 and Figure 7), supporting our proposal. In addition, we confirmed our previous observation that the MAC-CYP cells robustly suppressed experimental colitis (Figure 8). Moreover, this current study further demonstrated that the engineered macrophages significantly upregulated the expression of CYP27B1 gene and its encoded protein (1α-hydroxylase) in the inflamed intestines, which provides an extra source of local 1,25(OH)_2_D synthesis (Figure 8). Hence, de novo synthesis of locally high 1,25(OH)_2_D concentrations appears to be an attractive approach to circumvent the limitation of bringing 1,25(OH)_2_D to the bedside of IBD patients.

In addition to the Lgr5^+^ ISCs, previous data demonstrated the presence of reserved ISCs [25,26,27,28,29]. However, the exact markers for the reserved stem cells are still being debated. These reserved ISCs appear essential for intestinal epithelial repair under conditions that damage Lgr5^+^ ISCs [25,29,58,59]. Interestingly, a recent study using human organoids suggests that 1,25(OH)_2_D enhances the expression of several markers associated with the reserved ISCs, e.g., MSI1 and Mex3A [60]. Our ongoing investigation will determine how the de novo synthesized locally high 1,25(OH)_2_D concentrations impact the reserved ISCs, further revealing the therapeutic role of this innovative approach.

This study also showed that intestinal inflammation elevated the expression of the local CYP27B1 gene that encodes the 1α-hydroxylase (Figure 8E, left panel), consistent with the findings in other experimental colitis models and human IBD patients [61,62]. Therefore, our data and others suggest that the inflammation-induced local elevation of 1α-hydroxylase expression in inflamed intestines is a feedback mechanism aiming to combat intestinal inflammation in IBD patients. However, such a self-regulatory mechanism is obviously not adequate to control inflammation. In contrast, our engineered macrophages, the MAC-CYP cells, further increased local 1α-hydroxylase gene and protein expression in the inflamed intestines (Figure 8F–H) and were able to bring local 1,25(OH)_2_D to levels sufficient to suppress the local inflammation. 

In our system, we observed that local VDR expression was increased in the inflamed intestines in mice induced for DSS colitis (Figure 8E, middle panel), which differed from other reports showing that VDR expression in inflamed intestines are reduced in human IBD patients and mice induced for colitis by TNBS (1,4,6-trinitrobenzene sulfonic acid) [61,62]. The reduced local VDR expression could render IBD patients insensitive to the MAC-CYP cell treatment. However, it is worth mentioning that the MAC-CYP cells also robustly suppressed TNBS-induced colitis, as shown in our recently published data [7]. Therefore, our data strongly indicate that the MAC-CYP cells can overcome the local imbalance of vitamin D metabolism and effectively execute 1,25(OH)_2_D functions in the inflamed intestines of IBD patients.

In addition, our data and others demonstrate that intestinal inflammation does not increase the 1,25(OH)_2_D-degrading enzyme CYP24a1 (Figure 8E, right panel) [61]. Meanwhile, the MAC-CYP cell treatment significantly augmented the CYP24a1 expression (Figure 8E, right panel). Since CYP24a1 is a highly sensitive marker of active 1,25(OH)_2_D function [63], our data further reinforce that the MAC-CYP cells can effectively execute 1,25(OH)_2_D function locally in the inflamed intestines. Despite this elevated CYP24a1 expression, the MAC-CYP cell treatment could overcome this enhanced local 1,25(OH)_2_D catabolic mechanism and effectively suppress colitis.

So far, data from our laboratory and others have demonstrated many potentially beneficial effects of locally high 1,25(OH)_2_D concentrations for IBD patients without the currently known major adverse effect (i.e., hypercalcemia). Such beneficial effects may include enhancement of intestinal epithelial repair, as demonstrated in this study, suppression of inflammation [12,13,14,15,16,17,18,19,20], protection of the intestinal epithelial barrier [21,22,23,24], and blocking of IBD-associated cancer development [64]. However, since 1,25(OH)_2_D has broad biological functions, it remains unknown whether locally high 1,25(OH)_2_D concentrations may produce other currently unidentified effects on health. We will investigate this further in our future studies.

## 5. Conclusions

In this study, using in vitro high-purity Lgr5^+^ ISC lines and a mouse model, in which Lgr5^+^ ISCs and their progeny can be specifically tracked, we present evidence that 1,25(OH)_2_D at high concentrations enhances Lgr5^+^ ISC functions in vitro. In vivo, our data show that inflammation- and gut-homing macrophages engineered to overexpress the 1,25(OH)_2_D-synthesizing enzyme (1α-hydroxylase encoded by the CYP27B1 gene), MAC-CYP cells, increase local expression of the 1α-hydroxylase in inflamed intestines, robustly suppress established experimental colitis, and promote Lgr5^+^ ISC differentiation. Hence, our data further support the notion that de novo synthesis of locally high 1,25(OH)_2_D concentrations is a promising therapy for IBD.

## Figures and Tables

**Figure 1 cells-13-01465-f001:**
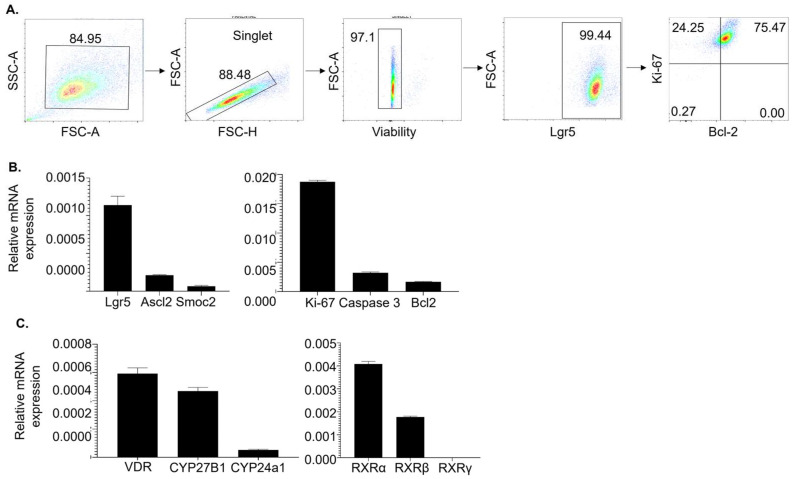
Lgr5^+^ ISC lines express stemness markers and molecules necessary for 1,25(OH)_2_D signaling. (**A**) Lgr5^+^ ISC lines were generated as described in the Materials and Methods Section and analyzed by flow cytometer. The data are representative flow cytometry plots showing the gating strategy and the expression of stem cell marker Lgr5, proliferation marker Ki-67, and anti-apoptotic marker Bcl-2. (**B**,**C**) The Lgr5^+^ ISC lines were examined by real-time RT-qPCR for expressing stem cell markers (Lgr5, Ascl2, and Smoc2) and molecules related to proliferation (Ki-67) and apoptosis (Caspase 3 and Bcl-2) (**B**); molecules related to vitamin D metabolism (VDR, CYP27B1, CYP24a1, RXRα, RXRβ, and RXRγ) (**C**). Bar represents the mean ± standard error of the mean (SEM) (n = 3).

**Figure 2 cells-13-01465-f002:**
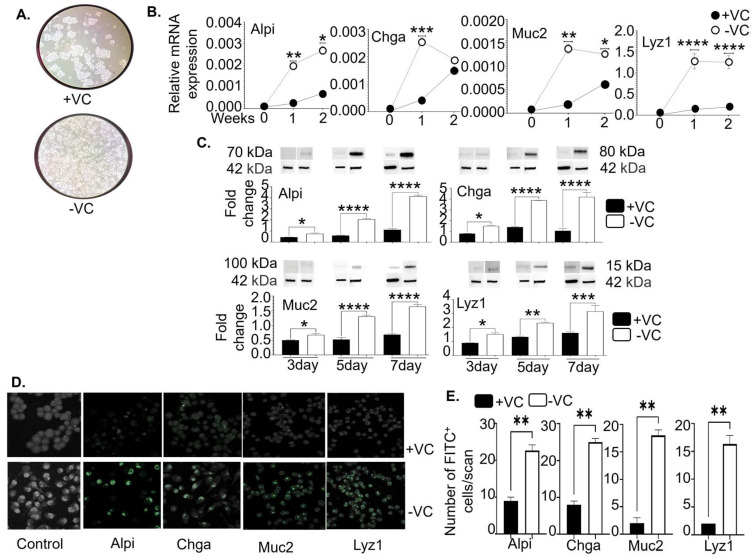
Lgr5^+^ ISC lines upregulate the expression of the markers for mature intestinal epithelial cells after removal of the chemical inhibitors (valproic acid and CHIR99021). The Lgr5^+^ ISC lines were cultured in the presence (+VC) or absence (−VC) of the chemical inhibitors. At different times, the cells were examined. (**A**) The cells were imaged one week later under a 10× brightfield microscope. (**B**) The cells were analyzed at baseline, week 1, and week 2 by real-time RT-qPCR for mRNA gene expression (normalized to GAPDH) of the markers for enterocytes (Alpi), enteroendocrine cells (Chga), Goblet cells (Muc2), and Paneth cells (Lyz1). * *p* < 0.05, ** *p* < 0.01, *** *p* < 0.001, and **** *p* < 0.0001 (n = 3). Non-parametric paired *t*-test. (**C**) The cells were studied by Western blot on days 3, 5, and 7. The data show fold changes in the protein expression (relative to β-actin [42 KDa]) for Alpi (70 KDa), Chga (80 KDa), Muc2 (100 KDa), and Lyz1 (15 KDa). Bar represents the mean ± standard error of the mean (SEM). * *p* < 0.05, ** *p* < 0.01, *** *p* < 0.001, and **** *p* < 0.0001. Ordinary one-way ANOVA (n = 3). (**D**) The cells were investigated by immunocytochemistry one week later. Representative images show positive staining (FITC) for Alpi, Chga, Muc2, and Lyz1. (**E**) Cumulative data of “D” show the average number of FITC^+^ cells per scan field of randomly selected three microscopic fields. ** *p* < 0.01. Two-way ANOVA (n = 3).

**Figure 3 cells-13-01465-f003:**
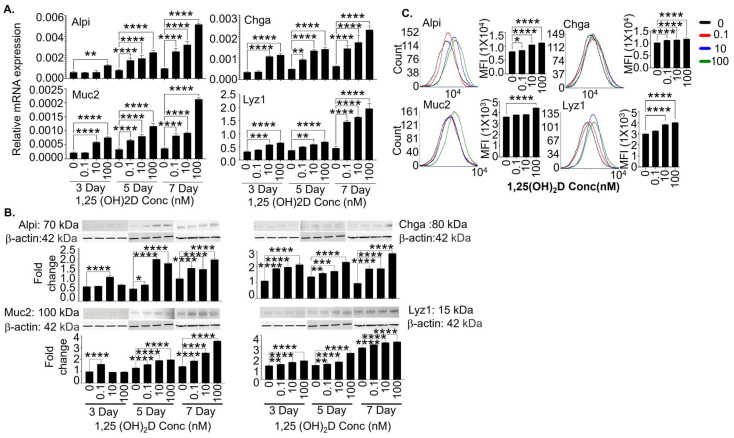
The 1,25(OH)_2_D at concentrations higher than physiological levels strongly enhances the gene and protein expression of mature intestinal epithelial markers in the Lgr5^+^ ISC lines, even in the presence of valproic acid + CHIR99021. (**A**) The Lgr5^+^ ISC lines were cultured in the presence of the chemical inhibitors and treated with different 1,25(OH)_2_D concentrations (0, 0.1, 10, and 100 nM). On days 3, 5, and 7, the cells were analyzed by real-time RT-qPCR. The data show relative mRNA expression (normalized to GAPDH) of the markers for enterocytes (Alpi), enteroendocrine cells (Chga), Goblet cells (Muc2), and Paneth cells (Lyz1). (**B**) The Lgr5^+^ ISC lines were cultured and treated as shown in “A” and examined by Western blot. The data show protein expression (relative to β-actin) of Alpi, Chga, Muc2, and Lyz1. (**C**) The Lgr5^+^ ISC lines were cultured and treated with different concentrations of 1,25(OH)_2_D (0, 0.1, 10, and 100 nM) in the presence of the chemical inhibitors. On day 7, the cells were studied by flow cytometer. The data show representative flow cytometer plots and cumulative data of Alpi, Chga, Muc2, and Lyz1 expression. Bar represents the mean ± standard error of the mean (SEM). * *p* < 0.05, ** *p* < 0.01, *** *p* < 0.001, and **** *p* < 0.0001. Ordinary one-way ANOVA (n = 3).

**Figure 4 cells-13-01465-f004:**
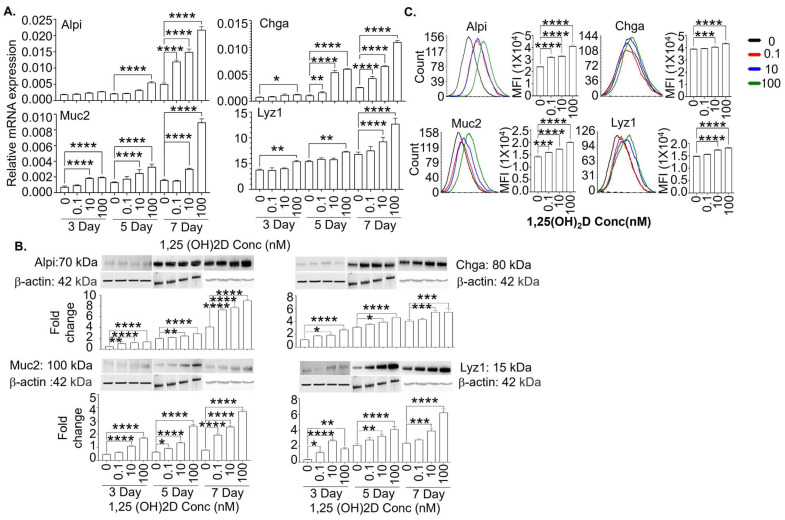
The 1,25(OH)_2_D at concentrations higher than the physiological levels robustly enhances mature intestinal epithelial markers’ gene and protein expression in the Lgr5^+^ ISC lines when the chemical inhibitors (valproic acid + CHIR99021) are removed to initiate differentiation. (**A**) The Lgr5^+^ ISC lines were cultured in the absence of the chemical inhibitors to initiate differentiation and, at the same time, treated with different 1,25(OH)_2_D concentrations (0, 0.1, 10, and 100 nM). The cells were analyzed by real-time RT-qPCR on days 3, 5, and 7. The data show relative mRNA expression (normalized to GAPDH) of the markers for enterocytes (Alpi), enteroendocrine cells (Chga), Goblet cells (Muc2), and Paneth cells (Lyz1). (**B**) The Lgr5^+^ ISC lines were cultured and treated as shown in (**A**). On days 3, 5, and 7, the cells were examined by Western blot. The data show representative images and cumulative data of Alpi, Chga, Muc2, and Lyz1 protein expression (relative to β-actin). (**C**) The Lgr5^+^ ISC lines were cultured and treated as shown in (**A**). The cells were studied by flow cytometer on day 7. Representative flow cytometer plots and cumulative data show Alpi, Chga, Muc2, and Lyz1 expression. Bar represents the mean ± standard error of the mean (SEM) (n = 3). Statistical analysis was performed by ordinary one-way ANOVA, * *p* < 0.05, ** *p* < 0.01, *** *p* < 0.001, and **** *p* < 0.0001.

**Figure 5 cells-13-01465-f005:**
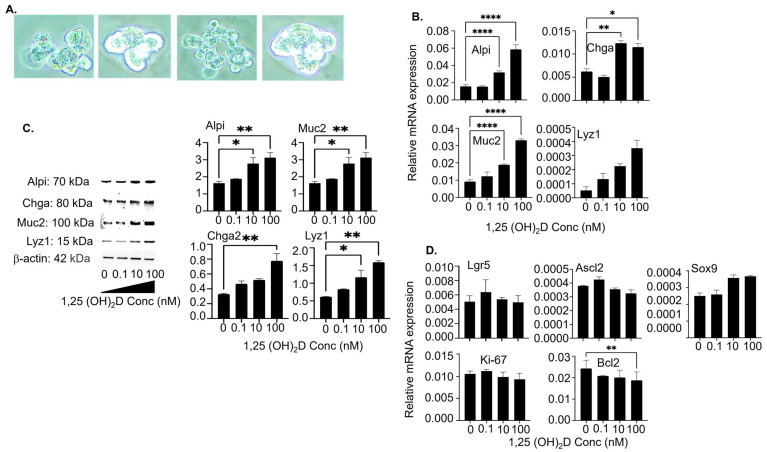
The 1,25(OH)_2_D at concentrations higher than physiological levels enhances mature intestinal epithelial markers’ gene and protein expression in colon organoids. Colon organoids were grown in Matrigel in the presence of 0, 0.1, 10, and 100 nM 1,25(OH)_2_D. (**A**) Representative 20× brightfield images of the organoids are shown. (**B**) The data show relative mRNA expression of the markers for mature intestinal epithelia (Alpi, Chga, Muc2, and Lyz1) at day 7. (**C**) The colon organoids were examined by Western blot on day 7. The left panel shows representative images of Alpi, Chga, Muc2, and Lyz1 protein expression. The right panel shows cumulative data. (**D**) The data show relative mRNA expression of the markers for stemness (Lgr5, Ascl2, and Sox9), proliferation (Ki-67), and anti-apoptosis (Bcl-2). The mRNA and protein expression was normalized to the housekeeping gene GAPDH and β-actin, respectively. Bar represents the mean ± standard error of the mean (SEM) (n = 3). Statistical analysis was performed by ordinary one-way ANOVA, * *p* < 0.05, ** *p* < 0.01, and **** *p* < 0.0001.

**Figure 6 cells-13-01465-f006:**
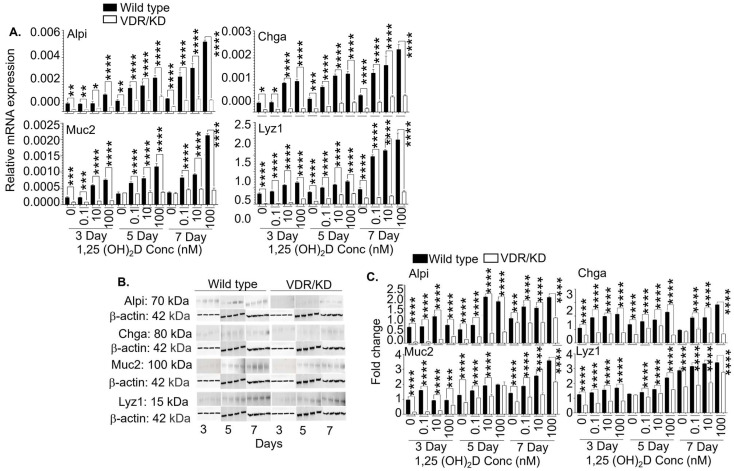
The effects of 1,25(OH)_2_D on Lgr5^+^ ISCs in the presence of chemical inhibitors depend on VDR. (**A**) VDR in the Lgr5^+^ ISC lines was knocked down as described in the Materials and Methods Section. The wild-type and VDR knockdown (VDR/KD) Lgr5^+^ ISC lines were treated with different 1,25(OH)_2_D concentrations (0, 0.1, 10, and 100 nM). The cells were analyzed by real-time RT-qPCR on days 3, 5, and 7 following the treatment. The data show relative mRNA expression of Alpi, Chga, Muc2, and Lyz1, normalized to the housekeeping gene GAPDH. (**B**) The wild-type and VDR knockdown Lgr5^+^ ISC lines were treated as described in (**A**). The cells were examined by Western blot on days 3, 5, and 7 following the treatment. Representative images show Alpi, Chga, Muc2, Lyz1, and β-actin protein expression. (**C**) Cumulative data of (**B**). Bar represents the mean ± standard error of the mean (SEM) (n = 3). Statistical analysis was performed by ordinary one-way ANOVA, * *p* < 0.05, ** *p* < 0.01, *** *p* < 0.001, and **** *p* < 0.0001.

**Figure 7 cells-13-01465-f007:**
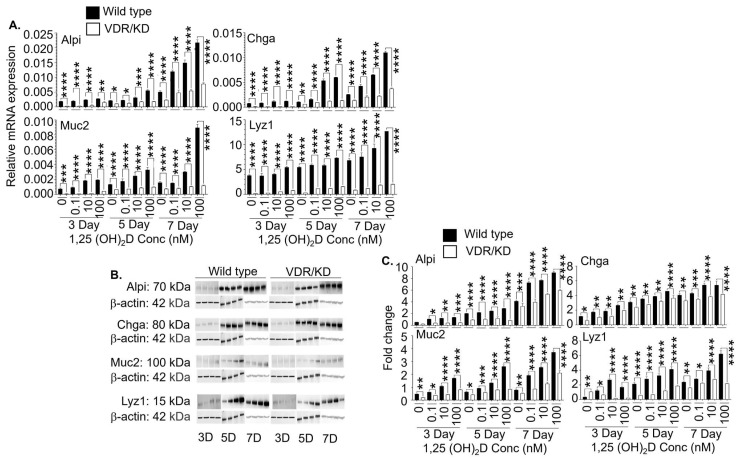
The effects of 1,25(OH)_2_D on Lgr5^+^ ISCs without chemical inhibitors depend on VDR. (**A**) The chemical inhibitors were removed in the wild-type and VDR knockdown Lgr5^+^ ISC lines to initiate differentiation. The cells were simultaneously treated with different concentrations of 1,25(OH)_2_D (0, 0.1, 10, and 100 nM) and analyzed by real-time RT-qPCR on days 3, 5, and 7. The data show relative mRNA expression of Alpi, Chga, Muc2, and Lyz1, normalized to the housekeeping gene GAPDH. (**B**) The wild-type and VDR knockdown Lgr5^+^ ISC lines were initiated for differentiation and treated as described in (**A**). On days 3, 5, and 7 following the treatment, the cells were examined by Western blot. Representative images show Alpi, Chga, Muc2, Lyz1, and β-actin protein expression. (**C**) Cumulative data of (**B**). Bar represents the mean ± standard error of the mean (SEM) (n = 3). Statistical analysis was performed by ordinary one-way ANOVA, * *p* < 0.05, ** *p* < 0.01, *** *p* < 0.001, and **** *p* < 0.0001.

**Figure 8 cells-13-01465-f008:**
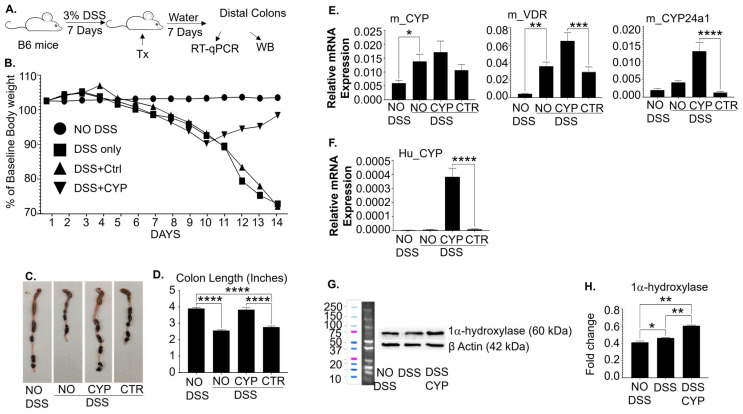
Peritoneal injection of MAC-CYP cells robustly suppresses experimental colitis and modulates local vitamin D metabolism. (**A**) Experimental design. (**B**) As described in the Materials and Methods Section, B6 mice were induced for DSS colitis. On day 7, the mice were divided into three groups, and each group intraperitoneally received one of the following treatments (Tx): no treatment (DSS only), 2 × 10^6^ MAC-Ctr cells/mouse (DSS+Ctr), and 2 × 10^6^ MAC-CYP cells/mouse (DSS+CYP). A group of healthy mice was also included as a control (No DSS). The mice were then monitored daily for survival and body weight. The data show the daily percentage of baseline body weight over the observation period, and no mice died during the period. (**C**) The mice were examined for distal colon length seven days after the treatment. Representative images are shown. (**D**) Average lengths of distal colons from (**C**) are shown. (**E**) The distal colons were studied by real-time RT-qPCR. The data show relative mRNA expression of mouse CYP27B1 (m_CYP), mouse VDR (m_VDR), and mouse CYP24a1 (m_CYP24a1). (**F**) The data show human CYP27B1 (Hu_CYP) mRNA expression levels normalized to the housekeeping gene GAPDH. Bar represents the mean ± standard error of the mean (SEM) (n = 3). (**G**) The distal colons were analyzed for 1α-hydroxylase protein expression by Western blot. Representative images show 1α-hydroxylase and β-actin protein expression. (**H**) Cumulative data of “G”. Statistical analysis was performed by ordinary one-way ANOVA, * *p* < 0.05, ** *p* < 0.01, *** *p* < 0.001, and **** *p*< 0.0001.

**Figure 9 cells-13-01465-f009:**
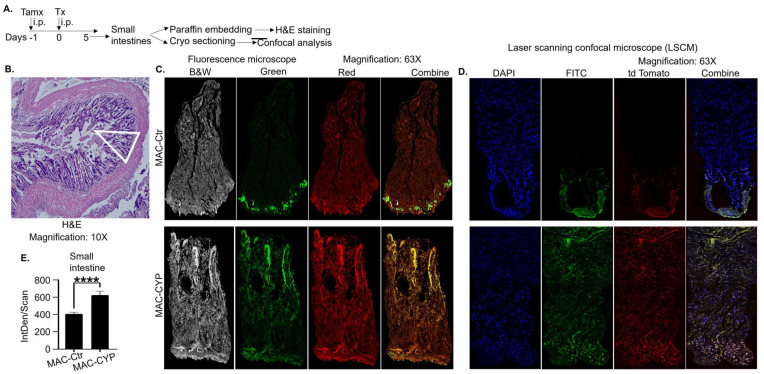
Peritoneal injection of MAC CYP cells enhanced Lgr5^+^ ISC differentiation in vivo. Lgr5GFP AI mice were generated by breeding the cre-inducible Lgr5-EGFP-IRES-creERT2 (Lgr5 GFP) mice with the cre-reporter B6.Cg-Gt(ROSA)26Sor tm9(CAG-tdTomato)Hze/J (Ai9 or Ai) mice. The mice were intraperitoneally injected with tamoxifen (75 mg/kg BW) (Tamx) to label the Lgr5^+^ ISCs with tdTomato permanently. On the second day, the mice received 2 × 10^6^ of either MAC-CYP or MAC Ctr cells. Five days later, small intestines were collected and processed via paraffin embedding for H&E staining or cryo sectioning for confocal analysis. (**A**) Experimental design. (**B**) Representative images of small intestine tissues in H&E stain at 10× magnification. (**C**) Representative fluorescence microscope images at 63× magnification where GFP (green) identifies Lgr5^+^ ISCs, and tdTomato (red) identifies both Lgr5^+^ ISCs and their differentiated progeny. (**D**) Representative laser scanning confocal microscope (LSCM) images at 63× magnification where blue color identifies nucleus staining by DAPI, green color identifies Lgr5^+^ ISCs, red color identifies both Lgr5^+^ ISCs and their differentiated progeny, and the combined image shows the mixture of all three, i.e., blue, green, and red colors. (**E**) The data show the integrated intensities (IntDen/Scan) of tdTomato^+^ cells (red, crypts are excluded) per scan from three randomly chosen microscopic fields, quantified by ImageJ software (Version 1.54j). Bar represents the mean ± standard error of the mean (SEM) (n = 5). Statistical analysis was performed by one way ANOVA. **** *p* < 0.0001. MAC Ctr: macrophages transduced with a control lentiviral vector, and MAC-CYP: macrophages transduced with lenti-CYP.

## Data Availability

The original contributions presented in the study are included in the article and Appendix A. Further inquiries can be directed to the corresponding author.

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
