# Peer review of "1,25-Dihydroxyvitamin D Enhances the Regenerative Function of Lgr5+ Intestinal Stem Cells In Vitro and In Vivo"

_cells, 2024, doi:10.3390/cells13171465_

Round 1

Reviewer 1 Report

Comments and Suggestions for Authors

General:

The study by Nisar Ali Shaikh et.al. with the title “1,25-Dihydroxyvitamin D enhances the regenerative function of Lgr5+ intestinal stem cells in vitro and in vivo” is important in this field, but I have, however, the following comments.

1.     A little unclear what distinguishes High-purity Lgr5single ISC culture from Organoid culture? Are the cells taken from the same Crypt single-cell isolation step?

2.     You study Alpi, Chga, Muc2 and Lyz1, and in the results section you report the results from this in figure 2-7 without developing the results in the discussion. I wish you could take up these findings and discuss what they stand for in the discussion.

3.     However, you discuss the expression of 1a-hydroxylase in the discussion on several occasions and refer to figure 8. But I cannot see that in figure 8 you are studying the expression of 1a-hydroxylase. This also makes your conclusion unclear. This needs to be clarified.

4.     As it seems that a high concentration of vitamin D is required for effect, I would like to have a discussion about advantages versus disadvantages regarding this in the discussion.

5.     The text to figure 2 can be shortened.

6.     It appears that the expression of B-actin also increases after 5-7 days compared to 3 days (Figure 3 and 6). How du you explain it and can you really use B-actin for normalization as it is also affected? Have you tried any other house-keeping protein? E.g. GAPDH that you use in PCR?

7.     Why are the western blot images tilted so much in the figure? The original images don´t.

8.     The images are generally very small.

9.     Why aren't the organoids grown on membranes that make the cells to form a natural "cell layer" with an apical and a basolateral side and with the possibility to stimulate the lumen side separately from the blood side?

10.  Shortened the text in figure 5.

11.  What gender was the mice?

Author Response

  1. Question No.1: A little unclear what distinguishes High-purity Lgr5+single ISC culture from Organoid culture? Are the cells taken from the same Crypt single-cell isolation step?

Yes, both cultures started with the same crypt single-cell suspensions.

To perform high-purity Lgr5+ single ISC lines, we reconstituted the crypt cells in the X-vivo 15 serum-free medium supplemented with EGF, Noggin, and R-Spondin 1 (collectively called ENR), along with valproic acid (V) and CHIR99021 (C). The presence of V+C suppresses Lgr5+ ISC differentiation and, therefore, promotes the proliferation of the Lgr5+ ISCs, eventually leading to high-purity Lgr5+ ISC lines.

For organoid cultures, the same crypt single cells were embedded in Matrigel and cultured in the X-vivo serum-free medium containing ENR only (without V+C). In the absence of V+C, the Lgr5+ ISCs continuously proliferate and differentiate. Because they are in the Matrigel, the cells organize themselves into the intestinal structure, including crypts that contain mainly Lgr5+ ISCs and villi that contain mature intestinal epithelial cells.

  1. Question No. 2: You study Alpi, Chga, Muc2 and Lyz1, and in the results section you report the results from this in figure 2-7 without developing the results in the discussion. I wish you could take up these findings and discuss what they stand for in the discussion.

We have now revised the discussion to describe what these markers stand for (line 618-624), which is copied below:

“In this study, we employed two strategies to assess the role of 1,25(OH)2D in Lgr5+ ISCs. In one strategy, we generated high-purity Lgr5+ ISC lines and assessed their differentiation in vitro into mature intestinal epithelial cells using well-defined markers, i.e., Alpi for mature enterocytes,39 Chga for enteroendocrine cells,40 Muc2 for Goblet cells,41 and Lyz1 for Paneth cells.42 Our data demonstrated that 1,25(OH)2D at concentrations significantly higher than physiological levels enhanced Lgr5+ ISC differentiation in vitro (Fig 1-7).”

  1. Question No. 3: However, you discuss the expression of 1a-hydroxylase in the discussion on several occasions and refer to figure 8. But I cannot see that in figure 8 you are studying the expression of 1a-hydroxylase. This also makes your conclusion unclear. This needs to be clarified.

Sorry for the confusion. MAC-CYP cells carry the overexpressed CYP27B1 gene, which encodes the 1a-hydroxylase protein. Figures 8E and F measure the CYP27B1 gene expression, whereas Figures 8G and H measure 1a-hydroxylase protein expression. We have now changed the “CYP27B1” in Figure 8G and “CYP” in Figure 8H to 1a-hydroxylase.   

  1. Question No. 4: As it seems that a high concentration of vitamin D is required for effect, I would like to have a discussion about advantages versus disadvantages regarding this in the discussion.

             We have now added a discussion, which is copied below:

             “So far, data from our laboratory and others have demonstrated many potentially beneficial effects of locally high 1,25(OH)2D concentrations for IBD patients without the currently known major adverse effect (i.e., hypercalcemia). Such beneficial effects may include enhancement of intestinal epithelial repair, as demonstrated in this study, suppression of inflammation12-20, protection of the intestinal epithelial barrier21-24, and blocking of IBD-associated cancer development.63 However, since 1,25(OH)2D has broad biological functions, it remains unknown whether locally high 1,25(OH)2D concentrations may produce other currently unidentified effects on health. We will investigate this further in our future studies.”

  1. Question No. 5: The text to figure 2 can be shortened.

We have shortened the Figure 2 legend.

  1. Question No. 6: It appears that the expression of B-actin also increases after 5-7 days compared to 3 days (Figure 3 and 6). How du you explain it and can you really use B-actin for normalization as it is also affected? Have you tried any other house-keeping protein? E.g. GAPDH that you use in PCR?

We are sorry for the confusion. Some images were distorted when shrunk to fit into the figures. The distortion in Figures 3 and 6 appears more apparent because protein concentrations were relatively low when the two inhibitors (valproic acid + CHIR99021) were present. We have now corrected those images based on the original images we submitted. Although we tried to maintain the same loading quantities for all the samples in one gel, some samples may vary because of low protein concentrations. Overall, in our experimental system, we did not see beta-action fluctuations as long as we could maintain the same loading volumes.

  1. Question No. 7: Why are the western blot images tilted so much in the figure? The original images don´t.The images are generally very small.

            As stated by the reviewer, there were no issues with the original images submitted. We tried to include all related data in one figure to support a conclusion. However, a large amount of data also entails smaller images. In addition, some images were distorted when being processed into one figure. We have now made the necessary corrections and improved the image quality.

  1. Question No. 8: Why aren't the organoids grown on membranes that make the cells to form a natural "cell layer" with an apical and a basolateral side and with the possibility to stimulate the lumen side separately from the blood side?

The crypt-villus structure in the intestines involves the migration of the Lgr5+ ISCs upwards during differentiation. Therefore, it is three-dimensional (3D) and can be conveniently visualized using organoid culture. Although the 3D crypt-villus structure is not the focus of this study, it will be necessary for our future investigations.  A membrane-based culture system may not recapitulate the 3D structure. However, it is worth determining how the Lgr5+ ISCs in the ENR medium behave in the membrane-based system, which will be evaluated in our future studies.

  1. Question No. 9: Shortened the text in Figure 5.

We have shortened the Figure 5 legend per the reviewer’s suggestion.

  1. Question No. 10: What gender were the mice?

We used male mice in this study, which is now included in the Materials and Methods.

Reviewer 2 Report

Comments and Suggestions for Authors

The manuscript entitled “1,25-Dihydroxyvitamin D Enhances the Regenerative Function of Lgr5+ Intestinal Stem Cells in Vitro and in Vivo” demonstrates that vitamin D might play a role in supporting gut health and potentially aiding Lgr5+ stem cell function in IBD through VDR receptor on their surface. The role of Lgr5+ stem cells in IBD recovery is very important as depletion and/or non-functional Lgr5+ stem cells contribute to the chronic inflammation and tissue damage. However, exact mechanism of action (MOA) was unknown. The paper explains a possible MOA.

The manuscript is well-written displaying results/information in targeted and well-designed fashion after thorough analysis. The available research is clearly presented, discussed, and the conclusion is supported by the evidence presented. The paper is very interesting and fit for the publication.

Author Response

We appreciate the reviewer’s time and effort in evaluating our manuscript. We hope this study can help other investigators in this field to understand further 1,25(OH)2D’s role in intestinal stem cells.

Reviewer 3 Report

Comments and Suggestions for Authors

In this manuscript by Nisar Ali Shaikh et al, the authors aimed to demonstrate the effects of 1,25(OH)2 vitamin D on the differentiation of Lgr5+ intestinal stem cells (ISCs). The authors developed a purification method for high-purity Lgr5+ ISCs and found that these cells responded to 1,25(OH)2 vitamin D treatment that resulted in increased differentiation. The authors further elucidated that delivery of macrophages overexpressing CYP27B1 enhanced Lgr5+ ISC differentiation and alleviated DSS-induced colitis in mice. While this is a thorough study with therapeutic potential, there are a number of concerns that need to be addressed before this manuscript is in a publishable fashion. Specific comments are as follows:

  1. The authors tested the effect of removing valproic acid and CHIR99021 on Lgr5+ ISCs and found the this increased the expression of mature epithelial markers. This verifies the results by Yin et al that these two chemicals help maintain the homogeneity of Lgr5+ ISC cultures. However, the following experiments with 1,25(OH)2 vitamin D treatment were also conducted with or without  valproic acid/CHIR99021 (Figure 3&4 and 6&7). This makes little sense as there are no data or references showing these two would change the expression or activity of VDR. It is advised to reorganize these data and emphasize only on vitamin D.
  2. In Figure 2E and its figure legend, the term "GFP+" seems awkward as these ISCs were isolated from regular B6 mice and subjected to immunocytochemical staining?
  3. In Figure S2, are there significant differences between groups for Ascl2 and Sox9 expression?
  4. In Figures 6 and 7, the term "VDRKO" should be rephrased as these were gene knocked down cells.
  5. It is understandable to have functional readouts such as MAC-CYP treatment for experimental colitis (Figure 8). However, there should at least be some direct evidence connecting the therapeutic effects with Lgr5+ ISCs.
  6. In the results of Figure 9, the authors stated 'suggesting enhanced migration and differentiation of the Lgr5+ ISCs' (Line 599). How was enhanced differentiation shown as there were only data from EGFP and tdTomato fluorescence?
  7. As in 6), in both Figures 9 and S5, there seems to be an increase in tdTomato fluorescence. Does that mean increased cell number of Lgr5+ cells? In addition, why is there a decrease in GFP intensity in the DC-CYP group while the tdTomato intensity is increased? Don't all the red cells also have green?
  8. The preparation of dendritic cells is suggested to be included in the methods.

Author Response

Comments and Suggestions for Authors

In this manuscript by Nisar Ali Shaikh et al, the authors aimed to demonstrate the effects of 1,25(OH)2 vitamin D on the differentiation of Lgr5+ intestinal stem cells (ISCs). The authors developed a purification method for high-purity Lgr5+ ISCs and found that these cells responded to 1,25(OH)2 vitamin D treatment that resulted in increased differentiation. The authors further elucidated that delivery of macrophages overexpressing CYP27B1 enhanced Lgr5+ ISC differentiation and alleviated DSS-induced colitis in mice. While this is a thorough study with therapeutic potential, there are a number of concerns that need to be addressed before this manuscript is in a publishable fashion. Specific comments are as follows:

  1. The authors tested the effect of removing valproic acid and CHIR99021 on Lgr5+ ISCs and found the this increased the expression of mature epithelial markers. This verifies the results by Yin et al that these two chemicals help maintain the homogeneity of Lgr5+ ISC cultures. However, the following experiments with 1,25(OH)2 vitamin D treatment were also conducted with or without valproic acid/CHIR99021 (Figure 3&4 and 6&7). This makes little sense as there are no data or references showing these two would change the expression or activity of VDR. It is advised to reorganize these data and emphasize only on vitamin D.

Response: Physiologically, Lgr5+ ISCs are present in the intestinal crypts, and they start to differentiate into mature intestinal epithelial cells when exiting the crypts. It is worth mentioning that most Lgr5+ ISCs lose Lgr5 expression after they exit the crypts. Therefore, most mature intestinal epithelial cells do not express Lgr5. However, based on our observations and others, Lgr5 expression is occasionally maintained in some mature intestinal epithelial cells, especially during MAC-CYP treatment, as shown in Figure 9.

            In the culture system described in this study, the addition of valproic acid (V) and CHIR99021 (C) maintains the stemness of the Lgr5+ ISCs, which mimics the microenvironment in the intestinal crypts. Removing VC initiates the differentiation of the Lgr5+ ISCs, which recapitulates the microenvironment outside the intestinal crypts.

            In vivo, 1,25(OH)2D could potentially act on both microenvironments (the crypts and the villi), affecting Lgr5+ stem cells in the crypts and their differentiation process outside the crypts, which was why we investigated how 1,25(OH)2D impacted the Lgr5+ ISC lines under both conditions (with and without VC). We apologize for the confusion and have added more explanations to the text, as copied below. We hope we have now clarified this issue.

-------------------------------

Line 370-372:

Hence, the Lgr5+ ISC lines maintained in the ENR-VC condition are bona fide stem cells, which closely mimic the intestinal crypt microenvironment.

Line 408-411:

Hence, the culture condition without VC recapitulates the intestinal microenvironment outside the crypts. In conclusion, the high-purity Lgr5+ cell lines are one invaluable tool for understanding 1,25(OH)2D’s effects on Lgr5+ ISCs and their differentiation.

Line 433-436:

We hence cultured the high-purity Lgr5+ ISC lines with different 1,25(OH)2D concentrations (0, 0.1, 10, and 100 nM) in the continuous presence of valproic acid and CHIR99021. The purpose was to cast light on how 1,25(OH)2D impacts Lgr5+ ISCs inside the crypts.

Line465-466:

This data is consistent with Lgr5 downregulation after crypt stem cells migrate out of intestinal crypts.4

--------------------------------

  1. In Figure 2E and its figure legend, the term "GFP+" seems awkward as these ISCs were isolated from regular B6 mice and subjected to immunocytochemical staining?

Response: Thank you for pointing out this incorrect labeling. We have corrected it and used “FITC+ cells.”

  1. In Figure S2, are there significant differences between groups for Ascl2 and Sox9 expression?

Response: There are no significant differences between groups for Ascl2 and Sox9.

Throughout the manuscript, the significant differences are marked with asterisks; the absence of asterisks between the groups indicates that they are not significant.

  1. In Figures 6 and 7, the term "VDRKO" should be rephrased as these were gene knocked down cells.

Response: Sorry for the confusion. We have now replaced the term “VDRKO” with “VDR/KD” in Figures 6 and 7. 

  1. It is understandable to have functional readouts such as MAC-CYP treatment for experimental colitis (Figure 8). However, there should at least be some direct evidence connecting the therapeutic effects with Lgr5+ ISCs.

Response: The main focus of this manuscript was to determine how 1,25(OH)2D affects stemness and differentiation of Lgr5+ ISCs. In vitro, we add different 1,25(OH)2D concentrations in high-purity Lgr5+ ISC lines under stemness (with VC) and differentiation (without VC) conditions. In vivo, we asked whether MAC-CYP cells could deliver locally high 1,25(OH)2D concentrations and how Lgr5+ ISC differentiation was affected. We used the colitis model mainly to emphasize that our MAC-CYP cells can create locally high 1,25(OH)2D concentrations in inflamed intestines (MAC-CYP cells have gut- and inflammation-homing properties).

We agree with the reviewer that understanding the contribution of Lgr5+ ISCs to colitis suppression is important. However, in addition to its differentiation-promoting role, 1,25(OH)2D has many other functions that may help suppress colitis, e.g., induction of regulatory T cells and protection of intestinal barrier integrity. To tease out these individual factors, we will need special mouse strains to knock out individual molecules separately. For example, we will need a unique mouse strain where Lgr5+ ISCs can be conditionally knocked out immediately before colitis induction (Germline Lgr5+ ISC knockout kills mice and cannot be used). For the above reasons, we have a separate project to address the roles of 1,25(OH)2D in different compartments (via MAC-CYP cells) and their contributions to colitis suppression using various unique mouse strains.

  1. In the results of Figure 9, the authors stated 'suggesting enhanced migration and differentiation of the Lgr5+ ISCs' (Line 599). How was enhanced differentiation shown as there were only data from EGFP and tdTomato fluorescence?

Response: In this experiment, we used heterozygous Lgr5GFP-AI mice, which were in-house bred by crossing cre-inducible Lgr5-EGFP-IRES-creERT2 (Lgr5-GFP) mice with the cre-reporter B6.Cg-Gt(ROSA)26Sor tm9(CAG-tdTomato)Hze/J (Ai9 or AI) mice.

               In the Lgr5GFP-AI mice, Lgr5 is only expressed in the stem cells located in the crypts and physiologically turned off in mature intestinal epithelial cells. Therefore, GFP (green color) is only present in the crypts normally (as shown in the mice treated with MAC-Ctr cells). However, we noticed that some mature intestinal epithelial cells outside the crypts still expressed Lgr5 when the mice were treated with MAC-CYP cells and displayed a green color.

               In contrast, tdTomato expression is not controlled by the Lgr5 promoter and is expressed in both crypt Lgr5+ ISCs and their differentiated progenies (mature epithelial cells) outside the crypts. Hence, tdTomato+ cells outside the crypts represent differentiated progenies of crypt Lgr5+ ISCs. More tdTomato+ cells outside the crypts represent enhanced migration and differentiation of crypt Lgr5+ ISCs.

We have added the following sentences to clarify this issue:

--------------------

Line 599-602:

Because most mature intestinal epithelial cells lose Lgr5 expression under physiological conditions, GFP is predominantly only located in the intestinal crypts. In contrast, tdTomato is independent of Lgr5 expression, and hence, red color is present in both intestinal crypts and villi.

----------------------

  1. As in 6), in both Figures 9 and S5, there seems to be an increase in tdTomato fluorescence. Does that mean increased cell number of Lgr5+ cells? In addition, why is there a decrease in GFP intensity in the DC-CYP group while the tdTomato intensity is increased? Don't all the red cells also have green?

Response: Please see our explanation in “6.”

  1. The preparation of dendritic cells is suggested to be included in the methods.

Response: We have now added the dendritic cell protocol in the Method section, which is copied below:

-------------------------------

Line 227-238:

Experiments using DC-CYP and DC-Ctr cells

DC.24 cells, kindly provided by Dr. Kenneth L. Rock,36 are a DC line generated from bone-marrow-derived DCs and have been shown to recapitulate DC functions in vivo closely.37,38 DC2.4 cells were cultured in X-VIVO 15 serum-free media (Lonza Biosciences, Cat#: 04-418Q) at 5% CO2 and 37°C. At 70% confluency, the DC 2.4 cells were transduced with lenti-CYP or lenti-Ctr (multiplicity of infection=40), as described above, to generate DC-CYP and DC-Ctr cells, respectively. 

            For the in vivo experiment, 2 x 106 DC-CYP or DC-Ctr cells/mouse in 100 ml PBS were administered intraperitoneally. Five days later, intestines were harvested and subjected to either paraffin embedding for H&E staining or cryo-sectioning for confocal analysis.

---------------------------------

Round 2

Reviewer 3 Report

Comments and Suggestions for Authors

The reviewer thanks the authors for appropriately addressing the concerns and has no further questions.